# A Multi-Path Compensation Method for Ranging in Wearable Ultrasonic Sensor Networks for Human Gait Analysis

**DOI:** 10.3390/s19061350

**Published:** 2019-03-18

**Authors:** Karalikkadan Ashhar, Mohammad Omar Khyam, Cheong Boon Soh

**Affiliations:** 1School of Electrical and Electronic Engineering, Nanyang Technological University, 50 Nanyang Ave, Singapore 639798, Singapore; ecbsoh@ntu.edu.sg; 2Department of Mechanical Engineering, Virginia Tech, Blacksburg, VA 24060, USA; mok@vt.edu

**Keywords:** multi-path compensation, correlation receiver, chirp compression, gait analysis

## Abstract

Gait analysis in unrestrained environments can be done with a single wearable ultrasonic sensor node on the lower limb and four fixed anchor nodes. The accuracy demanded by such systems is very high. Chirp signals can provide better ranging and localization performance in ultrasonic systems. However, we cannot neglect the multi-path effect in typical indoor environments for ultrasonic signals. The multi-path components closer to the line of sight component cannot be identified during correlation reception which leads to errors in the estimated range and which in turn affects the localization and tracking performance. We propose a novel method to reduce the multi-path effect in ultrasonic sensor networks in typical indoor environments. A gait analysis system with one mobile node attached to the lower limb was designed to test the performance of the proposed system during an indoor treadmill walking experiment. An optical motion capture system was used as a benchmark for the experiments. The proposed method gave better tracking accuracy compared to conventional coherent receivers. The static measurements gave 2.45 mm standard deviation compared to 10.45 mm using the classical approach. The RMSE between the ultrasonic gait analysis system and the reference system improved from 28.70 mm to 22.28 mm. The gait analysis system gave good performance for extraction of spatial and temporal parameters.

## 1. Introduction

Gait analysis is an important clinical tool to assess disorders due to neurovascular or musculoskeletal diseases and to study relations between gait and falls in the elderly population [1]. The gold standard system for human motion capture and gait analysis is an optical system with high-speed infrared cameras [2]. Optical motion tracking requires dedicated laboratories with controlled lighting conditions which makes this system unfit for unrestrained environments or a homely setup [3]. Non-traditional methods developed for human motion tracking include magnetic sensors [4], laser sensors [5,6], inertial measurement units [7,8], ultrawideband (UWB) ranging sensors [2,9], ultrasonic sensors [1,10] etc. Force sensing platforms or wearable force sensors can be used to the get ground reaction forces, the center of pressure and the joint moments [11]. Magnetic sensors are affected by the heterogeneity of the earth’s magnetic field and other ferromagnetic materials [12]. Laser sensors take a long time to take one sample of the whole scenario due to their scanning pattern [13]. Inertial sensors are affected by the drift during integration and fluctuating offset values and require sensor fusion to obtain accurate results [14]. UWB sensors demand high accuracy clock synchronization and high-cost ADCs to provide accurate ranging measurements [1]. Ultrasonic sensors can provide high accuracy time of flight (ToF) measurements without using expensive hardware as the wave travels much slower compared to electromagnetic waves and thus it can tolerate small errors in clock synchronization between the transmitting and receiving nodes [15]. A wireless portable and economic gait analysis system with only one ultrasonic marker attached to the lower limb was proposed by Qi et al. [14]. Here eight pulses of single tone frequency of 40 kHz were used for ranging. A net RMSE of 42.99 mm in 3-dimensional space was obtained compared to optical motion capture system [16]. A similar method was used for tracking both the lower limbs using spatial division multiple access in [10] where net RMSE for tracking each lower limb was less than 35 mm. In these methods, multipath components were removed by keeping an inhibit time after the reception. Time-of-Arrival (ToA) can be estimated in ranging sensors using different methods such as thresholding, cross-correlation with the originally transmitted signal or by comparing the phase of the received signal with the original. A phase-correlation method to improve the correlation performance was explained in [17], which requires higher SNR levels to provide satisfactory performance. The cross-correlation method is accurate and robust compared to other methods in a noisy environment [18]. As the noise increases, the performance of cross-correlation decreases if single-tone signals are used. Binary phase shift keying (BPSK) was used to code the transmitted signals in [19] to improve the cross-correlation performance. However, the performance degrades in time-varying fading channels [20]. A significant improvement in cross-correlation performance can be observed when linear frequency modulated signals are used [21,22]. A cross-correlation method with linear chirp signals can be used in the ultrasonic gait analysis system to improve the accuracy and robustness of the system in noisy environments [23]. The accuracy of cross-correlation increases with the bandwidth of the chirp signals [17]. Even though broadband ultrasonic sensors provide higher accuracy compared to narrowband ultrasonic sensors, these sensors come at a higher cost and operating voltage. Hence, we use narrowband piezoelectric transducers along with linear chirp signals for ranging in our experiments. In traditional correlation receivers, the time at which the peak of the correlation happens is used to estimate the ToF of the ultrasonic signal.

In a cluttered environment, the transmitted ultrasonic waves can travel in different paths to reach the receiver leading to multi-path interference. An earliest peak search method was implemented in [24] to remove multi-path interference, where the earliest peak of the correlation which is above the noise floor and which might not necessarily be the highest peak was considered as the correlation peak of the direct path. However, when the multi-path component lies closer to the line of sight (LOS) component so that the distance between the LOS and the Non-line of sight (NLOS) component is less than the range resolution of the signal, we cannot identify separate peaks in the correlation signal at the receiver. In this case, the peaks corresponding to LOS and the NLOS cannot be differentiated. A super-resolution technique based on MUSIC algorithm was implemented in [25,26,27]. The techniques based on MUSIC or matrix pencil [28] are computationally intensive as they involve the computation of each singular eigenvector and corresponding eigenvalue [27]. A phase-correlation method was implemented to get a sharp peak at the correlation output [17]. However, this method cannot be used in a noisy environment. Interpolation methods were used on the obtained correlation using narrowband chirp signals in [29,30] which cannot always provide accurate results and takes more time for analysis. In this paper a method to reduce the ranging errors caused by close multi-path interference is discussed.

We propose a novel multi-path compensation method for the ultrasonic indoor localization and gait analysis system. The contribution of this paper is the adaptation of the concept of half-peak detection from radio frequency systems [31] to indoor ultrasonic localization system and gait analysis and further extension of this idea. The idea of half-peak detection to reduce multi-path errors was never used in ultrasonic systems. We propose an earliest $\frac{1}{m}$ peak detection algorithm in ultrasonic localization system and use this for gait analysis. Initially, we do the simulation experiments followed by the ranging experiment and the gait analysis experiment with five healthy subjects and we compare the performance of the system with an optical motion tracking system.

The rest of the paper is organized as follows: Section 2 explains the proposed multi-path compensation method. Section 3 explains the simulations conducted and their results. Section 4 explains the setup used for the experiments with ultrasonic sensors and the gait analysis system. Results from the experiments, the performance of tracking and estimation of spatial and temporal parameters are mentioned in Section 5. Finally, conclusions from the work are explained in Section 6 with some suggestions for future work.

## 2. Proposed Method

In the proposed method, linear chirp signals are used for ranging. Chirp pulse compression is the process of transforming a long duration frequency modulated pulse into a narrow pulse with much higher amplitude. The pulse compression process can be considered as an application of matched filter system. Let g(t) be the impulse response of the transmitter coding filter. Then, the response of the matched filter at the receiver should be $g^{*}\left(-t\right)$, where $g^{*}\left(\right)$ represents the complex conjugate of g() and the output of the matched filter receiver can be given as shown in Equation (Equation 1).
(1)$$y\left(t\right)=\int_{-\infty }^{\infty }g\left(\tau \right)g^{*}\left(t-\tau \right)d\tau$$

In chirp compression process, consider g(t) as a linear chirp signal multiplied by a rectangular window function given by
(2)$$g\left(t\right)=rect\left(\frac{t}{T}\right)e^{2\pi j(f_{0}t+kt^{2}/2)}$$

From Equation (Equation 1), the output of chirp compression process, y(t) can be derived as
(3)$$y\left(t\right)=\sqrt{TB}\times \frac{sin(\pi Bt)}{\pi Bt}\times \sqrt{j}e^{j2\pi (f_{0}t-kt^{2}/2)}$$
where, *T* is the chirp duration, *B* is the chirp bandwidth and $f_{0}$ is the centre frequency. If we transmit a linear chirp signal with signal duration T seconds, the output, y(t) will be a waveform with sync characteristics with signal duration 2T seconds. In traditional ultrasonic ranging methods, the time at which the peak of y(t) happens is taken as an estimate of the ToF. The low cost, low power piezoelectric transducers comes with a narrow bandwidth around 40 kHz. We used MA40S4S/R transducers from MURATA Electronics which has a bandwidth of about 2 kHz. Hence, we selected the frequency range from 39 to 41 kHz. We selected linear chirp signals as they are robust in terms of Doppler sensitivity and frequency selective fading [32]. Non-linear chirp signals give better results when the time-bandwidth product is higher, which was 14 in our experiments. Also, non-linear chirps give higher side lobs in the presence of Doppler effect [33].

In the chirp compression process, the range resolution of an ultrasonic ranging sensor is the minimum separation of two targets ranging using the same signal that can be identified as separate targets from the received signals. The range resolution of the ultrasonic ranging in ideal case with a chirp signal multiplied by rectangular window function can be given by v/δf, where *v* is the speed of sound and δf is the bandwidth of chirp. In this study, we consider linear chirp signal from 39–41 kHz, thus δf=2 kHz and the range resolution was found to be 172 mm when the speed of sound was 345.3 m/s. If the extra path length of the multi-path component lies within 172 mm, which is quite likely to happen in a multi-path environment, there might not be two separate peaks and the resultant peak can be shifted in the time axis. The earliest peak detection algorithm explained in [24] fails here, leading to errors in the measured value. Let $T_{p}$ be the position of the peak value of correlation signal mentioned in Equation (Equation 1) and $P_{0}$ be the peak value. In a close multi-path environment, the position of the peak will be shifted to $T_{p}+X$. Where *X* represents the error. Now, instead of taking $P_{0}$ to calculate the ToF, we propose to take $\frac{P_{0}}{m}$ crossing point to the left of earliest peak instead of the actual peak of the absolute value of the correlation waveform to calculate the ToF. The earliest $\frac{1}{m}$th peak will be less affected by the close multi-path components as the multi-path components travel a higher distance than the LOS path and hence appears time-shifted in the cross-correlation of the received signal with the originally transmitted signal. Since the correlation output at the receiver is sync signal with the chances of corruption by the multi-path components higher at the descending part of the main peak as shown in Figure 1, the chances of shift in time domain at the earliest $\frac{1}{m}$ peak point (ascending part of the main peak) will be less. In this work, the value of *m* is chosen out of 2, 3 and 4 by conducting simulations with 10,000 iterations with different multi-path components each time. However, $\frac{P_{0}}{m}$ should be kept at a safe margin above the channel noise floor. We selected the value of *m* as 2, 3 and 4 for simulation experiments keeping *m* = 1 for conventional method and the best value was chosen for implementation in the proposed gait analysis system. The pictorial representation of the proposed method is shown in Figure 1. Since we are removing any constant offset which is present in the range measurement, we are concerned only about the relative distance between two range measurements of the same transmitter. Even though the existence of a close multi-path shifts the time at which the correlation peak happens, the earliest $\frac{1}{m}\times$ peak point will be less affected compared to the actual peak. The chirp signals can be multiplied with various kinds of window functions to increase the sensitivity and to reduce the amplitude of side lobes in the signal [34]. We test the performance of three different window functions (rectangular, hamming and hann) along with the proposed method. Let S(t) be the transmitted chirp signal multiplied by the window function, W(t) given by Equation (Equation 4).
(4)$$S\left(t\right)=W\left(t\right)\times exp\left\{j2\pi \left(f_{s}t+\frac{1}{2}\left(f_{e}-f_{s}\right)t^{2}\right)+\phi \right\}$$
where, $f_{s}$, $f_{e}$ and ϕ represents the starting frequency, the ending frequency and the initial phase respectively. The received signal without any multi-path components can be represented as
(5)$$\hat{S}\left(t\right)=A_{0}\left(h\left(t\right)*S\left(t\right)\right)$$
where, $A_{0}$, h(t) and * represents the amplitude after path loss, channel impulse response and the convolution operator respectively. Considering *M* multi-path components, the correlation of received signal can be given by Equation (Equation 6).
(6)$$C\left(t\right)=\hat{S}\left(t\right)⋆S\left(t\right)+\left(\sum_{m=1}^{M}A_{m}\hat{S}\left(t-\tau_{m}\right)+n\left(t\right)\right)⋆S\left(t\right)$$
where, ⋆, n(t), $A_{m}$ and $\tau_{m}$ represents the correlation operator, channel noise, the amplitude and the extra path travelled by the multi-path components respectively. In Equation (Equation 6), the actual ToA information from LOS path is embedded in the first term. The second term in Equation (Equation 6) is the multi-path noise. Our hypothesis is that the earliest point at which the value of C(t) crosses $\frac{1}{m}$ (where, m>1) of the peak of C(t) will be less affected by the multi-path components (the second term in Equation (Equation 6)) in time axis compared to the peak of C(t). This is because the multi-path components appear at a later time compared to the LOS path.

### Doppler Compensation in the Received Signal

Moving sensor nodes emitting ultrasonic waves create Doppler effect and this alters the actual range measurements. A Doppler compensation method with linear up- and down-chirp signals was explained in [23]. Even though linear chirp signals show higher Doppler-tolerance, range-Doppler coupling leads to an error which is proportional to the moving velocity. For down-chirp signals, the error due to Doppler effect will be the same in magnitude with different polarity to that of up-chirp signals. In our experiments, in each ranging cycle of 40 ms, ranging with up-chirp was conducted in the initial 20 ms and with down-chirp in the next 20 ms duration. The chirp duration was selected as 7 ms as we allocated 20 ms for up-chirp and 20 ms for down-chirp and out of 20 ms, 13 ms time was provided between two signals so that the echo will die out completely. Let $R_{1}$, $R_{2}$ and $R_{0}$ be the range estimated from the up-chirp, down-chirp and the actual range respectively. Then,
(7)$$\begin{array}{cc} R_{1} & =R_{0}+\dot{R}\times f_{0}T/B\\ R_{2} & =R_{0}-\dot{R}\times f_{0}T/B\\ R_{0} & =(R_{1}+R_{2})/2 \end{array}$$
where $\dot{R}$ represents the Doppler velocity of the moving target.

## 3. Simulation Results

A customized ranging environment with one transmitter and one receiver was simulated in Matlab to test the performance of the multi-path compensation technique. The amplitude of signal after attenuation in air was modeled as $A=A_{0}e^{-\gamma \times \delta d}$, where $A_{0}$ and δd are the originally transmitted amplitude and the distance traveled through the medium respectively. The value of attenuation constant, γ was selected as 0.17 Np/m. The ranging was done with a chirp signal with linearly increasing frequency from 39 to 41 kHz in 7 ms. In order to test the performance of other non-linear chirp signals, a simulation was conducted to find the performance of linear, logarithmic and quadratic chirp signals to estimate the Doppler-velocity using the method explained in Section 2.1 with 1000 iterations. The results show that the performance of the linear chirp was better than the logarithmic and quadratic chirp signals. A close multi-path component (NLOS1) with a reflection coefficient of 0.9 was provided at random positions so that the difference between the LOS and the NLOS was less than the range resolution which is 172 mm in this case. The relative distance between two LOS values which were set to 1000 mm and 2000 mm respectively was calculated to find the improvement in ranging accuracy using our proposed method and the classical peak detection method. A Monte Carlo simulation with 10,000 iterations was done to estimate the distance at each position with different multi-path reflections. We selected the time index at which the cross-correlation of the received signal with the original signal crosses $\frac{1}{m}$ of its earliest peak value for the first time as an estimate for the LOS path. The plots of the correlation waveform from the LOS, NLOS and the combined signal with *m* = 2, 3 and 4 points marked are shown in Figure 1.

The results obtained when the transmitted signals are multiplied by three different kinds of window functions are as shown in Table 1, where, $I_{\mathrm{av}}$, $I_{\max}$, $D_{\mathrm{av}}$ and $D_{\max}$ represent the average improvement, maximum improvement, average degradation, and maximum degradation in absolute error compared to peak detection respectively. Out of 10,000 iterations, N(I) and N(D) are the number of cases with improvement and degradation respectively and the ratio of average improvement to average degradation was calculated as $R=I_{\mathrm{av}}/D_{\mathrm{av}}$. In order to find the changes in the ratio, R as the value of *m* changes, we repeated the same simulation experiment with 10,000 iterations and a rectangular window function with 10 dB channel SNR for each value of *m* increasing from 1 to 4 in steps of 0.1 and the plot of R against *m* was shown in Figure 2. It can be observed that R slightly decreases after *m* = 1.9 as the value of *m* increases. We can also observe a sudden increase in *R* as *m* changes from 1 to 1.1.

We found that the “hann” and “hamming” windows showed better results in terms of the ratio, R for *m* = 3 and 4. This can be explained by lower side lobs obtained for the correlation output *y*(*t*). With the attenuated side lobs, the time shift by the multi-path component in the $\frac{1}{m}$ peak also decreases thus resulting in better R values. However, it should be noted that as the value of *m* increases, the estimate is getting closer to the noise floor and chances of errors and $D_{\max}$ also increase. Hence, optimum value of *m* depends on the peak value of cross correlation and noise power. It can also be observed that the maximum degradation of the accuracy is higher for “hann” and “hamming” windows.

The plot of average improvement and average degradation in the absolute error while ranging with linear chirp signals multiplied with rectangular window and a channel with 10 dB SNR is shown in Figure 3.

The Empirical Cumulative Distribution Function (ECDF) for the errors in range estimation from the proposed method and the conventional peak detection method was calculated for rectangular window function and *m* = 2 and plotted in Figure 4. The plot shows that the proposed method outperforms the conventional peak detection method in a high multi-path environment. Thus, we selected *m* = 2 and a rectangular window function at the transmitter side for our tests using ultrasonic sensors for further analysis.

## 4. Materials and Methods

Initially, we conduct ranging experiments with one static transmitter and receiver in an indoor environment followed by the gait analysis experiment with four fixed receivers and one mobile node. Murata MA40S4S and MA40S4R ultrasonic transducers are used for ranging with chirp signals. These sensors have a narrow bandwidth with 40 kHz center frequency. The transmitter node was designed on an STM32F4-discovery development board which has a form factor, 66 mm × 97 mm. However, in the future, this system can be printed on a flexible wearable band to make it easier and unobtrusive to use. The mobile node was powered by an 11.1v Li-Po battery and a 5v voltage regulator. So the driving voltage of the ultrasonic transmitter was 11.1v. No wired connections were limiting the movement of the subject as we used wireless clock synchronization between the mobile node and the anchor nodes using 2.4 GHz wireless modules. The attachment of the mobile node to the lower limb is as shown in the Figure 5. The chirp signals were stored in the memory of the development board and are sent periodically to the transmitters through digital to analog converters and a driver IC (SN754410) using direct memory access. At the receiver side, we used a bandpass filter, an amplifier and a data acquisition board. ADCs with the same resolution (which was 125 kilo-samples/s in our experiments) can replace the data acquisition board in the future.

The power consumption of the mobile node is important in wireless sensors nodes as the mobile nodes are battery powered. Here the main part of power consumption comes from the memory access and the driving of ultrasonic transducers. We used STM32F4-discovery board to develop the sensor node and SN754410 as the driver for MA40S4S transmitter. Murata MA40S4S transmitters have 120 dB Sound Pressure Level (SPL) at 40 kHz (where, 0 dB = 0.02 mPa). An 11.1v battery and IC-SN754410 drove these transmitters. At ambient temperature, the absolute maximum continuous power dissipation in the driver IC is 2.075 W from the datasheet. The power dissipation in the development board assuming dynamic run mode at 180 MHz clock frequency is 0.22 W. The total power consumption is 2.295 W. Hence, the total energy required for one ranging cycle is 0.0918 J. Here we neglected the power consumption in the voltage regulators. In the future, custom designed circuits can greatly reduce the power consumption.

An optical motion capture system with six high-speed infrared cameras were used as the reference for the experiments. The reference system was calibrated for static and dynamic conditions and the accuracy was found to be 0.36 ± 17 mm. Reflective markers were used for this system.

### 4.1. Ultrasonic Ranging Experiment

For ranging experiment we used both up-chirp and down-chirp and the average value was calculated using the proposed half peak method (*m* = 2) and classical peak detection method. The transmitted signals were multiplied by rectangular window functions. Measurements were taken with ten different positions of the transducers each for a low multi-path environment and a high multi-path environment. In the low multi-path environment, the transducers were placed about 750 mm above the ground with a line of sight and without any obstacles in the vicinity. In the high multi-path environment, three obstacles with flat surface were introduced near the transmission path to reflect the ultrasonic waves.

### 4.2. Gait Analysis System

A wireless gait analysis system was designed to test the proposed method. An ultrasonic mobile node was attached to the left lower limb of the subject as shown in Figure 5. For spherical localization, four static anchor nodes at the corners of a 250 × 200 mm rectangle were mounted on a vertical board and kept parallel to the sagittal plane at about 1000 mm from the treadmill as shown in Figure 6. The reflective markers of the motion capture system were attached beside the ultrasonic transmitters and the receivers and the offsets between the markers and ultrasonic transducers were adjusted during the post-processing. Five healthy subjects (four males and one female) with age in between 24 to 31 were recruited for the gait analysis study. Each subject was asked to do a treadmill walk for one minute for each walking speed of 1, 2 and 3 kph with both the ultrasonic mobile node and the camera marker attached to one lower limb. All participants gave their informed consent for inclusion before they participated in the study.

Doppler shift affects the performance of the system. We took the average of range estimated from linearly increasing and decreasing chirp signals to offset these errors as explained in Section 2.1 [23]. Each ranging cycle consists of two ranging each with 20 ms duration, first one with an up-chirp from 39 to 41 kHz in 7 ms and second one with a down-chirp from 41 to 39 kHz in 7 ms. A separation of 13 ms was thus provided between any transmitted signals. Thus the total update rate of the system was about 25 Hz. Both half peak detection method *m* = 2 and classical peak detection method were implemented for all the range measured from all the four ultrasonic anchor nodes in the gait analysis system. The state space equation for calculating the *x*, *y* and *z* coordinates from the measured distances is shown in the Equation (Equation 8).
(8)$$\begin{array}{c} X_{i}=AX_{i-1}+q_{i-1}\\ Y_{i}=g\left(X_{i}\right)+r_{i} \end{array}$$
where, *A* and $g(X_{i})$ are the state transition matrix and the non-linear output function given by Equation (Equation 9).
(9)$$A=\left[\begin{array}{cc} I & T\times I\\ 0 & T \end{array}\right],\;g\left(X_{i}\right)=\left[\begin{array}{c} \sqrt{{\left(x_{i}-x_{1}\right)}^{2}+{\left(y_{i}-y_{1}\right)}^{2}+{\left(z_{i}-z_{1}\right)}^{2}}\\ \sqrt{{\left(x_{i}-x_{2}\right)}^{2}+{\left(y_{i}-y_{2}\right)}^{2}+{\left(z_{i}-z_{2}\right)}^{2}}\\ \sqrt{{\left(x_{i}-x_{3}\right)}^{2}+{\left(y_{i}-y_{3}\right)}^{2}+{\left(z_{i}-z_{3}\right)}^{2}}\\ \sqrt{{\left(x_{i}-x_{4}\right)}^{2}+{\left(y_{i}-y_{4}\right)}^{2}+{\left(z_{i}-z_{4}\right)}^{2}} \end{array}\right]$$

Here, *I* is a 3 × 3 diagonal matrix and $x_{1,2,3,4}$, $y_{1,2,3,4}$ and $z_{1,2,3,4}$ represents the coordinates of the anchor nodes. The process noise and the measurement noise are given by $q_{i}$ and $r_{i}$ respectively which are designed as Gaussian function with noise co-variances *Q* and *R* respectively. Here, $R=diag(e^{2},e^{2},e^{2},e^{2})$ and *e* was set to 5 mm in our system. The state matrix, $X_{i}={\left[x_{i}\;y_{i}\;z_{i}\;\dot{x_{i}}\;\dot{y_{i}}\;\dot{z_{i}}\right]}^{T}$ and measurement matrix, $Y_{i}={\left[d_{1,i}\;d_{2,i}\;d_{3,i}\;d_{4,i}\right]}^{T}$ where $x_{i}$, $y_{i}$, $z_{i}$, $\dot{x_{i}}$, $\dot{y_{i}}$ and $\dot{z_{i}}$ represents the three dimensional coordinates and the velocities along the directions and $d_{1,i}$, $d_{2,i}$, $d_{3,i}$ and $d_{4,i}$ represents the measured distances from each anchor nodes at *i*th instance. An Unscented Kalman Filter (UKF) [35] along with an Unscented Raunch-Tung Striebel smoother [36] was implemented to obtain the three dimensional trajectory from the measured distances and the state space equation explained in Equation (Equation 8). All the trajectories were low pass filtered using a butter-worth filter with cut off frequency 10 Hz. The reader can refer [23] for further details of the filters applied.

### 4.3. Estimation of Gait Parameters

We estimated the spatial and temporal gait parameters from the 3D coordinates of one lower limb. The estimation of some gait parameters from the x and z coordinates of a single gait cycle is as shown in the Figure 7. The peaks in the x-coordinate can be assumed as the heel-strikes and the troughs in the x-coordinate can be assumed as the toe-off points [16]. The following parameters were extracted:

#### 4.3.1. Stride Time (ST)

Stride Time is defined as the time taken from one heel-strike to the next heel-strike of the same foot and is given by the time between two minimum points of the x-coordinate.
(10)$$ST\left(i\right)=Time\left(min_{i+1}\left(x\right)\right)-Time\left(min_{i}\left(x\right)\right)$$

#### 4.3.2. Stride Length (SL)

Stride length is defined as the distance covered in one stride and can be estimated as twice the difference between the maximum and the minimum point of x-coordinate.
(11)$$\begin{array}{cc} S(i) & \;=\;max_{i}\left(x\right)-min_{i}\left(x\right)\\ SL(i) & \;=\;2\times S(i) \end{array}$$

#### 4.3.3. Swing Time (SW)

Time taken from the toe off to next heel strike is defined as swing time.
(12)$$SW\left(i\right)=Time\left(max_{i}\left(x\right)\right)-Time\left(min_{i}\left(x\right)\right)$$

#### 4.3.4. Stance Time (STT)

Time taken from the heel strike to next toe off is defined as stance time.
(13)$$STT\left(i\right)=Time\left(min_{i+1}\left(x\right)\right)-Time\left(max_{i}\left(x\right)\right)$$

#### 4.3.5. Maximum Foot Clearance (MFC)

The foot elevation during swing phase is termed as foot clearance. Maximum foot clearance is the difference between the maximum and the minimum points of z-coordinate during one cycle.
(14)$$MFC\left(i\right)=max_{i}\left(z\right)-min_{i}\left(z\right)$$

## 5. Results and Discussion

Static measurements were taken with one transmitter and one receiver placed at 10 different positions each for a high multi-path and relatively low multi-path environments. The first measurement was subtracted from the remaining nine measurements to obtain the differential distances. The differential distance calculated from the proposed and the classical peak detection method was compared to the camera distance measurements to obtain the mean error. The standard deviation of distance calculated at each static position was also calculated. The results are provided in Table 2.

It can be observed that the we were able to get consistent and considerable improvement in the tracking accuracy using the proposed method. In both high and low multi-path conditions, the average standard deviation for static measurements decreased considerably from 10.44 mm in classical method to 2.45 mm in the proposed method. Figure 8 shows the static measurement reading obtained from the classical peak detection and the proposed half-peak detection methods after removing the offset between the measurements. It is evident that the proposed method provides stable measurements readings compared to the classical method.

The mean error and standard deviation of error (STD Error) and root mean square error (RMSE) between the ultrasonic system and the camera system are provided in Table 3. Any temporal delay between the foot trajectory from the ultrasonic system and the the camera system was removed before analysis. The net RMSE from the proposed method was found to be 22.28 mm compared to 28.70 mm from the classical peak detection method

### Comparison of Estimated Gait Parameters

To quantify the performance of the proposed system with respect to the reference system, the mean, the standard deviation and the RMSE between the parameters from both the systems during treadmill walk with walking speed 3 kph are calculated and listed in Table 4. The proposed system gave good results for estimation of stance time, swing time and maximum foot clearance. However, our system slightly over-estimated the stride length consistently for all the subjects. This can be solved by removing the constant offset value from all the measurements. A Wilcoxon rank sum test was conducted to test the hypothesis that the gait parameters estimated from the ultrasonic and camera system came from a continuous distribution with equal medians for each subject. The tests failed to reject the null hypothesis with *p* > 0.05 for all subjects. In the case of stride length, the constant trend was removed before testing the hypothesis.

## 6. Conclusions

A new method of multi-path tolerant ultrasonic ranging and localization was proposed. The simulation experiments with earliest $\frac{1}{m}$ value detection shows that *m* = 3 and 4 performed better in terms of ratio, R, when the transmitted wave is multiplied by a “hann” or a “hamming” window. However, the maximum value of degradation increased for these window functions and *m* values. We implemented the half peak detection method with *m* = 2 and a rectangular window function on chirp signals and ultrasonic transducers. The proposed half-peak detection was found to perform better than the classical peak detection for coherent receiver in ultrasonic ranging system with static transducers as well as the localization system for gait analysis with one moving transmitter. The gait analysis system with one ultrasonic marker attached to one lower limb was found to give better results with net RMSE 22.28 mm for the proposed method compared to 28.70 mm for the classical method. The system gave good results for the estimation of spatial and temporal parameters from the proposed system. In future, the effects of SNR and different values of *m* on the ranging performance can be studied. The use of multiple sensors to increase the capture volume or use of cone-reflector to make the ultrasonic signals omni-directional [37] are also potential research opportunities in the future.

## Figures and Tables

**Figure 1 sensors-19-01350-f001:**
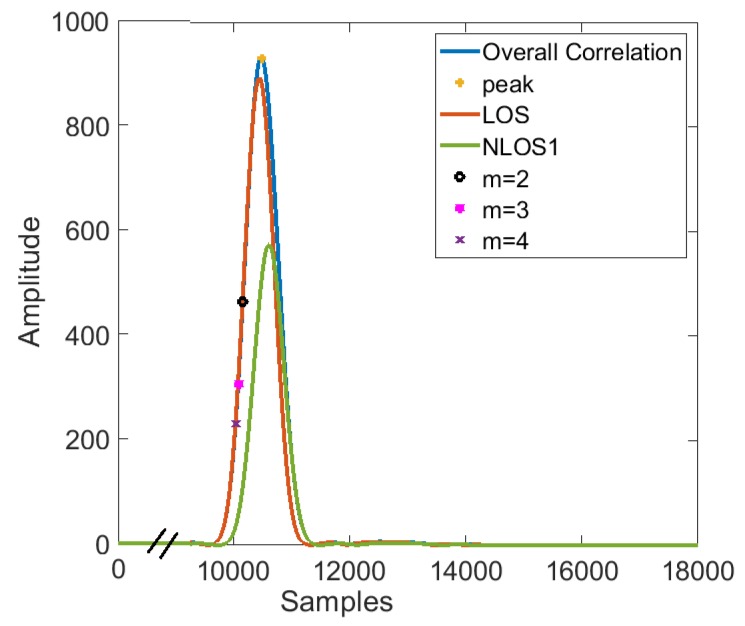
The envelope of correlation for LOS, one NLOS component (NLOS1) and the combined waveform.

**Figure 2 sensors-19-01350-f002:**
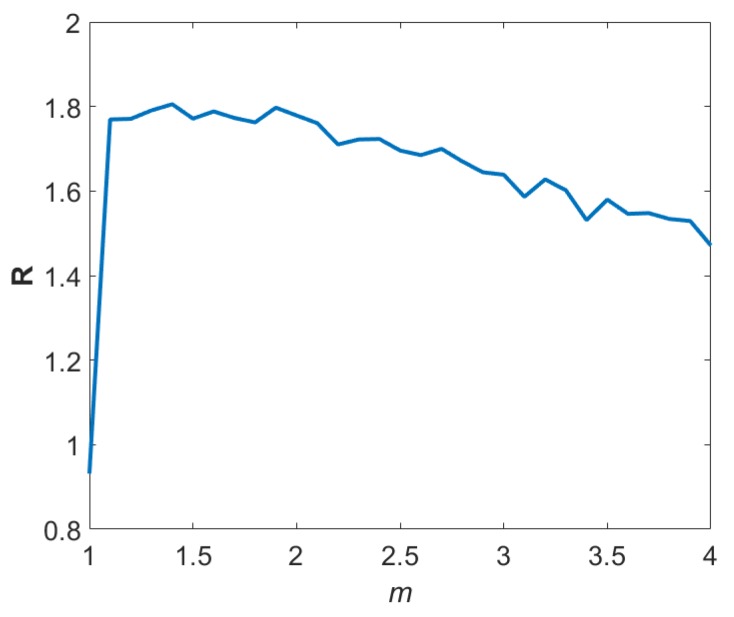
The plot showing variation of R as the value of *m* increases for a rectangular window function.

**Figure 3 sensors-19-01350-f003:**
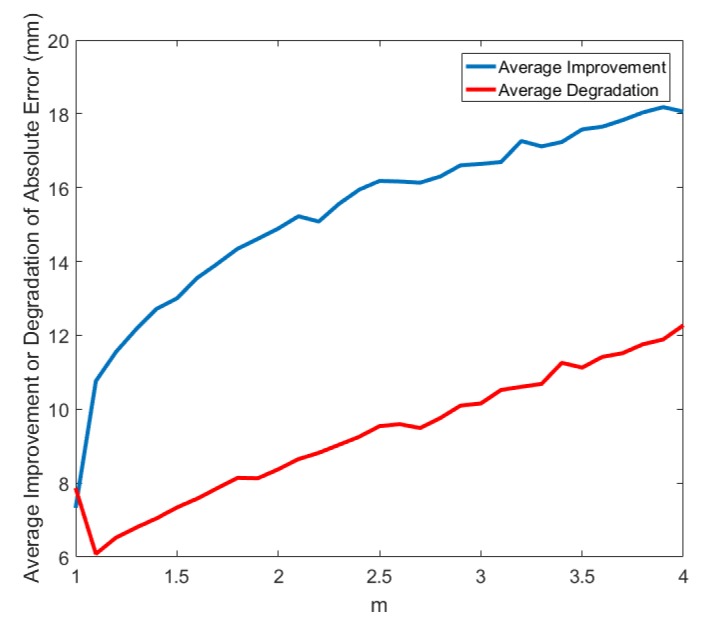
The plot showing average improvement and degradation in absolute error with the proposed method as the value of *m* changes.

**Figure 4 sensors-19-01350-f004:**
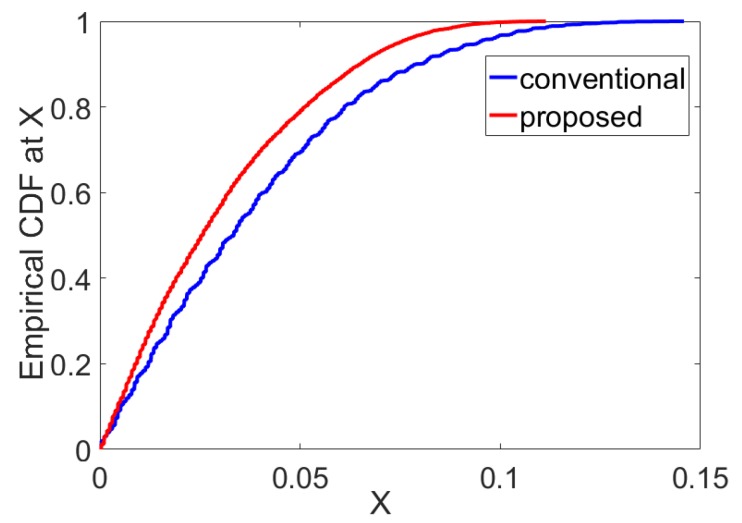
The Empirical CDF calculated for ranging error using half peak detection method and conventional peak detection method.

**Figure 5 sensors-19-01350-f005:**
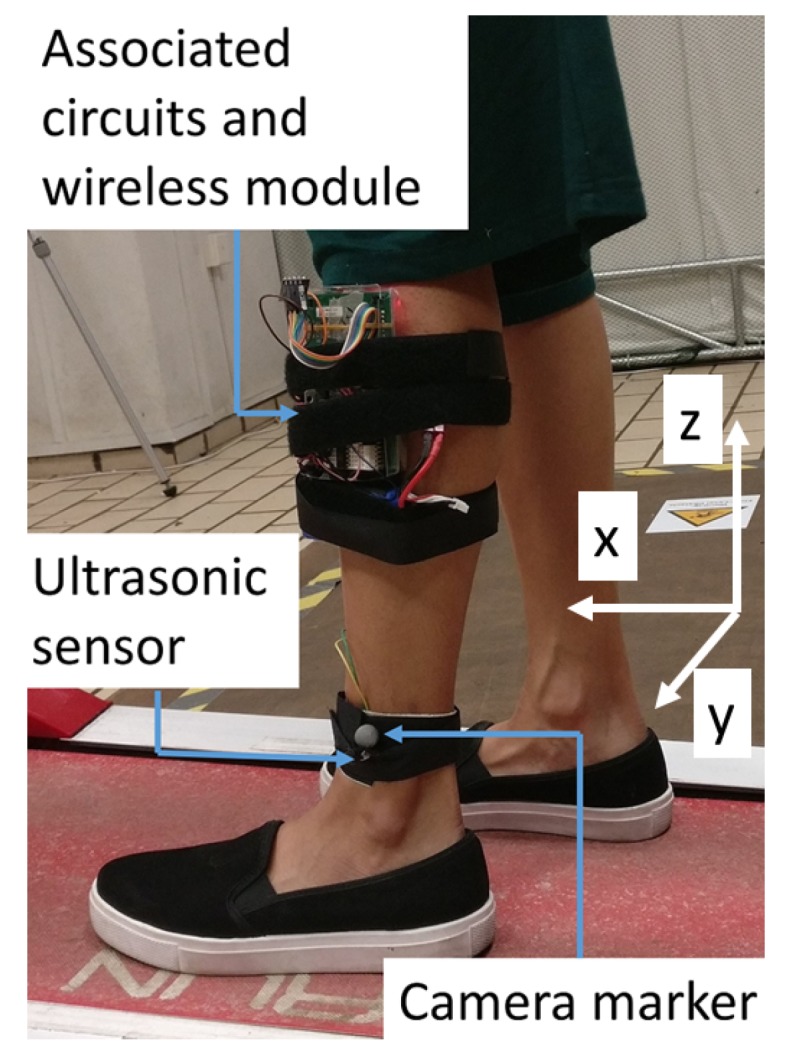
The mobile node and camera marker attached to left lower limb.

**Figure 6 sensors-19-01350-f006:**
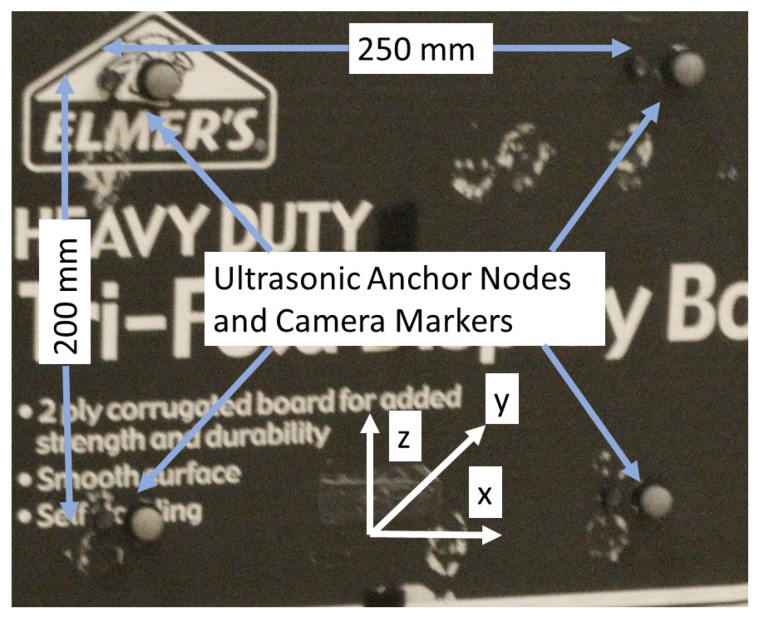
The anchor nodes attached on a vertical board along with the anchor nodes.

**Figure 7 sensors-19-01350-f007:**
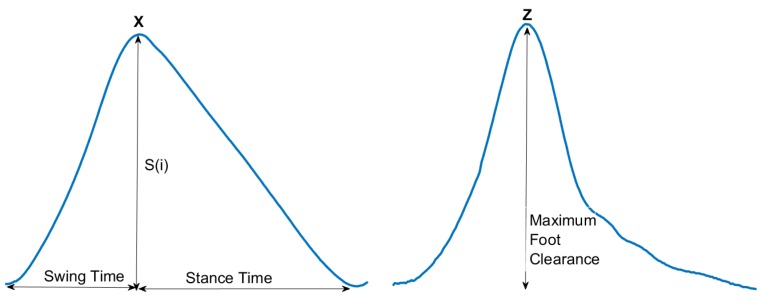
Estimation of gait parameters from a single gait cycle.

**Figure 8 sensors-19-01350-f008:**
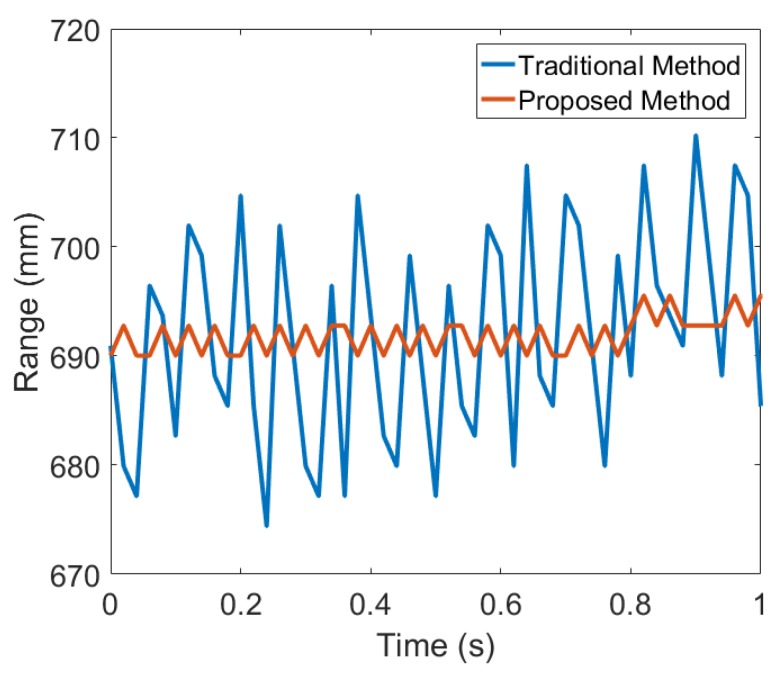
The static measurements obtained from the ultrasonic system using the classical and the proposed methods.

**Table 1 sensors-19-01350-t001:** Comparison of improvement and degradation of tracking performance of the system with earliest 1/*m* peak detection.

Window	*m*	$I_{\mathrm{av}}$ (mm)	$D_{\mathrm{av}}$ (mm)	$I_{\max}$ (mm)	$D_{\max}$ (mm)	N(I)	N(D)	R
Rectangular	2	14.8	8.4	55.8	35.9	6897	3046	1.76
	3	16.8	10.1	63.4	44.1	6826	3117	1.66
	4	18.2	12.4	68.3	55.2	6614	3336	1.47
Hann	2	21.6	10.9	104.8	116.5	7129	2826	1.98
	3	24.6	12.2	119.3	253.7	7135	2825	2.01
	4	26.5	14.2	118.6	446	7125	2842	1.87
Hamming	2	22.2	10	97.9	69.6	7365	2603	2.22
	3	25.9	11.3	120	129.6	7300	2660	2.29
	4	27.7	12.4	117.9	253.8	7310	2654	2.24

**Table 2 sensors-19-01350-t002:** Comparison of mean error in differential distance (ME) calculated from the proposed and the classical methods and the standard deviation in the static measurement using up-chirp ($S_{u}$) and down-chirp ($S_{d}$).

Method	High Multi-Path (mm)			Low Multipath (mm)		
	ME	$S_{u}$	$S_{d}$	ME	$S_{u}$	$S_{d}$
Proposed	26.63	2.66	2.61	7.70	2.27	2.26
Classical	44.10	10.34	10.41	22.24	10.68	10.32

**Table 3 sensors-19-01350-t003:** Comparison of error in estimated coordinates from proposed and classical methods.

Method	Proposed Method (mm)			Classical Method (mm)		
Coordinate	*x*	*y*	*z*	*x*	*y*	*z*
Mean Error	−0.94	−0.05	0.61	−0.97	0.24	0.19
STD Error	15.92	8.09	12.71	18.96	9.68	18.04
RMSE	16.04	8.14	12.92	19.32	9.79	18.40

**Table 4 sensors-19-01350-t004:** Comparison of mean and standard deviation (STD) of gait parameters estimated from the ultrasonic system and the reference camera system.

	Subject		1	2	3	4	5
Stride Length (mm)	Camera	Mean	995.74	898.76	1003.62	1204.28	932.1
		STD	68.95	93.12	135.36	160.69	125.93
	Ultrasound	Mean	1083.44	969.01	1069.37	1303.94	1016.46
		STD	71.67	107.43	144.05	173.94	143.88
	RMSE		88.68	75.83	69.99	102.47	91.13
Stance Time (ms)	Camera	Mean	802.08	648.93	720.27	835.92	645.72
		STD	35.34	59.98	77.79	44.83	35.44
	Ultrasound	Mean	798.12	642.95	714.64	833.9	631.98
		STD	36.52	63.97	84.06	40.27	38.86
	RMSE		16.4	15.79	15.19	10.61	19.33
Swing Time (ms)	Camera	Mean	521.46	402.78	459.91	483.33	431.76
		STD	13.19	21.57	25.29	26.82	19.03
	Ultrasound	Mean	525	408.76	465.76	484.48	445.04
		STD	16.77	23.49	25.74	32.02	19.88
	RMSE		15.65	15.45	16.78	12.95	17.27
Max. Foot Clearance (mm)	Camera	Mean	140.24	107.49	125.63	130.61	146.2
		STD	6.05	10.87	10.82	9.89	14.11
	Ultrasound	Mean	143.55	103.56	121.11	143.56	135.36
		STD	9.42	13.7	15.29	13.67	16.94
	RMSE		9.12	10.42	11.58	16.47	15.19

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
