# Peer review of "A Multi-Path Compensation Method for Ranging in Wearable Ultrasonic Sensor Networks for Human Gait Analysis"

_sensors, 2019, doi:10.3390/s19061350_

Round 1

Reviewer 1 Report

The authors have presented a method for multipath compensation for ranging in Ultrasonic sensor network. The proposed approach is interesting, but the presentation needs to be improved. To be specific,

What is the motivation for replacing P0 by P0/m. How to choose the value of m? Please clarify the correlation between m and the peak value of cross-correlation and noise power.

The impact of Doppler shift should be further explained. The equations should be discussed in the problem formulation, not within the experimental results.

Why does the R value improve when hann or hamming window is used?

More results need to be included to capture a wide range of scenarios. 

Author Response

The authors have presented a method for multipath compensation for ranging in Ultrasonic sensor
network. The proposed approach is interesting, but the presentation needs to be improved. To be
specific,

1.       What is the motivation for replacing P0 by P0/m. How to choose the value of m? Please clarify the correlation between m and the peak value of cross-correlation and noise power.

Response: Thank you for your comments. Our hypothesis in this work was that the earliest (1/m)th peak will be less affected by the close multipath components as the multipath components travel a higher distance than the LOS path and hence appears time shifted in the cross-correlation of the received signal with the original transmitted signal. Since the correlation output at the receiver is sync signal with the chances of corruption by the multipath components higher at the descending part of the main peak as shown in Figure-1, the chances of shift in time domain at the earliest (1/m) peak point (ascending part of main peak) will be less.

In this work, the value of m is chosen out of 2, 3 and 4 by conducting simulations with 10000 iterations with different multipath components each time. However, P0/m should be kept at a safe margin above the channel noise floor. It should be noted that the performance improvement or degradation also depends on the position of the multipath component and hence we cannot generalize the optimum value of m.

A few sentences are added in the proposed method in page number-3&4 to make it clear.

In order to find the changes in the ratio, R as the value of m changes, we repeated the same simulation experiment with 10000 iterations and a rectangular window function with 10 dB channel SNR for each value of m increasing from 1 to 4 in steps of 0.1 and the plot of R against m was shown in Fig 2. It can be observed that R slightly decreases after m=1.9 as the value of m increases. We can also observe a sudden increase in R as m changes from 1 to 1.1.

The simulation results were added in page-5.

2.       The impact of Doppler shift should be further explained. The equations should be discussed in the problem formulation, not within the experimental results.

Response: A method for Doppler correction was explained in our previous work [1] and was cited in the main text. In order to make things clear, a subsection-2.1 was added in the proposed method to explain the Doppler correction methods in page number-4:

Doppler compensation in the received signals: Moving sensor nodes emitting ultrasonic waves create Doppler effect and this alters the actual range measurements. A Doppler compensation method with linear up- and down-chirp signals was explained in [1]. Even though linear chirp signals are highly tolerant to Doppler effect, range-Doppler coupling leads to an error which is proportional to the moving velocity. For down-chirp signals, the error due to Doppler effect will be same in magnitude with different polarity to that of up-chirp signals. In our experiments, in each ranging cycle of 40 ms, ranging with up-chirp was conducted in the initial 20 ms and with down-chirp in the later 20 ms duration. Let R1, R2 and R0 be the range estimated from the up-chirp, down-chirp and the actual range respectively. Then,

Where,, ,  and  are the Doppler velocity, centre frequency, chirp duration and the chirp bandwidth respectively.

[1] Ashhar K, Khyam M, Soh C, Kong K. A Doppler-Tolerant Ultrasonic Multiple Access Localization System for Human Gait Analysis. Sensors. 2018 Aug;18(8):2447.

3.       Why does the R value improve when hann or hamming window is used?

Response: By using hann or hamming window, lower side lobs can be obtained for the correlation output y(t). Hence, with the attenuated side lobs, the time shift by the multi-path component in the 1/m peak also decreases thus resulting in better R values. However, it should be noted that the maximum degradation was found to increase with the application of hann or hamming windows.

A line was added in the main text in page-5 to explain this.

4.       More results need to be included to capture a wide range of scenarios.

Response: We added the plots of average improvement and degradation in absolute error as the value of m increases (Fig. 3). We also added more results from the gait analysis part and spatial and temporal parameter estimation in the manuscript.

Reviewer 2 Report

This paper proposed a  multi-path compensation method for the ultrasonic indoor localization and

gait analysis system. It is surprising that the paper didn't cover at all the gait analysis part. What gait parameters did you derive from your system is unclear. My second concern is how realistic to deploy the receivers and transmitter in practice...  

Author Response

We thank the editor and the reviewers for their constructive comments. In our revised manuscript, we have addressed all the received comments. In the following, we indicate for each individual comment how we have dealt with it. Note that reference numbers mentioned in the responses refer to the revised manuscript. We have highlighted major changes in the uploaded manuscript. The Response to the comments are attached.

Reviewer 3 Report

The paper elaborates on a multi-path compensation method for the ultrasonic indoor localization and gait analysis system. The work covers a timely topic and it is well written. I have some minor comments as follows:

1-   The novelty aspect of the paper is in question. There are numerous works in this subject area and authors should explicitly state their contributions in light of the existing literature.

2-   Why do authors consider a linear chrip in analysis?  How is the system performance affected if other techniques such as an exponential chirp is used? What is the rationale for increasing frequency from 39 to 41 kHz in 7 ms?

3-   As the studied wearable sensor pack (WSP) could be used for assessment of foot trajectory and healthcare purposes, could you provide some information regarding the energy level of WSP in dB/ Joule?

Author Response

We thank the editor and the reviewers for their constructive comments. In our revised manuscript, we have addressed all the received comments. In the following, we indicate for each individual comment how we have dealt with it. Note that reference numbers mentioned in the responses refer to the revised manuscript. We have highlighted major changes in the uploaded manuscript. The response to comments are attached

Round 2

Reviewer 1 Report

The authors have addressed my comments adequately. There is a typo in line 98 (page 3), 'lobs' should be 'lobes'.

Reviewer 2 Report

I am happy to recommend this paper for publication.